# SyMetric: Measuring the Quality of Learnt Hamiltonian Dynamics Inferred from Vision

**Irina Higgins**
DeepMind
London
irinah@deepmind.com

**Peter Wirnsberger**
DeepMind
London
pewi@deepmind.com

**Andrew Jaegle**
DeepMind
London
drewjaegle@deepmind.com

**Aleksandar Botev**
DeepMind
London
botev@deepmind.com

## Abstract

A recently proposed class of models attempts to learn latent dynamics from high-dimensional observations, like images, using priors informed by Hamiltonian mechanics. While these models have important potential applications in areas like robotics or autonomous driving, there is currently no good way to evaluate their performance: existing methods primarily rely on image reconstruction quality, which does not always reflect the quality of the learnt latent dynamics. In this work, we empirically highlight the problems with the existing measures and develop a set of new measures, including a binary indicator of whether the underlying Hamiltonian dynamics have been faithfully captured, which we call Symplecticity Metric or *SyMetric*. Our measures take advantage of the known properties of Hamiltonian dynamics and are more discriminative of the model's ability to capture the underlying dynamics than reconstruction error. Using SyMetric, we identify a set of architectural choices that significantly improve the performance of a previously proposed model for inferring latent dynamics from pixels, the Hamiltonian Generative Network (HGN). Unlike the original HGN, the new HGN++ is able to discover an interpretable phase space with physically meaningful latents on some datasets. Furthermore, it is stable for significantly longer rollouts on a diverse range of 13 datasets, producing rollouts of essentially infinite length both forward and backwards in time with no degradation in quality on a subset of the datasets.[1]

## 1 Introduction

If you want to understand how the world will change, a good place to start is with simple physical principles, such as the laws of motion. One way to achieve good performance in predicting dynamics forward and backward in time is by building a model using the Hamiltonian formalism from classical mechanics. This formalism aims to model physical systems which conserve energy. Apart from real physical problems, other important dynamical systems can also be modelled using this formulation, including GAN optimisation [35, 38, 34, 36], multi-agent learning in non-transitive zero-sum games [5, 6, 53], or the transformations induced by flows in generative models [41]. While many naturally observed dynamics do not preserve energy – for example, energy is often added or dissipated – they can

---

[1]The code for reproducing all results is available on https://github.com/deepmind/deepmind-research/tree/master/physics_inspired_models.

still be modelled using the Hamiltonian formalism by augmenting it with additional terms capturing the deviations from the energy-conserved state [37, 23, 43, 55, 56].

Recently, a body of work has emerged that brings these well-established principles of modelling dynamics from physics – such as the conservation of energy, and numerical formulations from the theory of differential equations – to neural network architectures [49, 20, 9, 3, 58, 8, 4, 31, 50, 43, 56, 12, 17, 24, 14, 32, 11, 45, 57, 54, 48, 21]. Most of these papers, however, learn dynamics directly from an abstract state, such as the positions and momenta of the physical objects.

This may be a suitable choice for some applications, for example in molecular simulations, but in situations where inference at test time has to be made from high-dimensional observations, like images – a setup common to many robotics, reinforcement learning or autonomous driving challenges – it is important to be able to have models that can both correctly infer the abstract state space and faithfully capture the underlying Hamiltonian dynamics without access to the ground truth state space information at inference time. Currently only a handful of methods exist for learning dynamics with physical priors from pixels [49, 9, 45, 3, 58]. These models learn in a completely unsupervised manner using pixel observations of the dynamics, however, currently there is no good way to know whether these models have indeed managed to recover the underlying dynamics faithfully through learning.

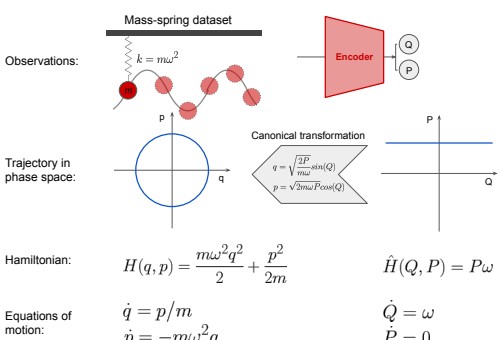

Figure 1: Illustration of a canonical transformation between the Cartesian phase space $(q,p)$ and the action-angle $(Q,P)$ representations for the mass-spring Hamiltonian system. Both coordinate spaces preserve the underlying dynamics.

In this paper we empirically demonstrate that reconstruction based measures typically used in the field are poor indicators of how well the model has learnt the underlying dynamics and propose a better method for evaluating such models. In summary, our contributions are as follows:

1. We introduce performance measures for identifying whether a model trained with the Hamiltonian prior has managed to capture the true dynamics of an energy-conserving system when learning from high-dimensional observations.

2. We use these measures to identify a set of hyperparameters and architectural modifications that significantly improves the performance of Hamiltonian Generative Networks (HGN) [49], an existing state of the art model for recovering Hamiltonian dynamics from pixel observations, both in terms of long time-scale predictions, and interpretability of the learnt latent space.

## 2   Preliminaries

**Hamiltonian dynamics**   The Hamiltonian formalism is a mathematical formulation of Newton's equations of motion for describing energy-conservative dynamics (see [25, 29]). It describes the continuous time evolution of a system in an abstract phase space with state $s = (q,p) \in \mathbb{R}^{2n}$, where $q \in \mathbb{R}^n$ is a vector of positions, and $p \in \mathbb{R}^n$ is the corresponding vector of momenta. The time evolution of the system in phase space is then given by the Hamiltonian equations of motion:

$$\dot{q} = \frac{\partial \mathcal{H}}{\partial p}, \quad \dot{p} = -\frac{\partial \mathcal{H}}{\partial q} \tag{1}$$

where the Hamiltonian $\mathcal{H} : \mathbb{R}^{2n} \to \mathbb{R}$ maps the state $s = (q,p)$ to a scalar representing the energy of the system. Based on the above equations of motion, the energy conservation property is easy to see, since $\frac{dH}{dt} = \frac{\partial H}{\partial q}\dot{q} + \frac{\partial H}{\partial p}\dot{p} = 0$.

**Hamiltonian dynamics as a symplectic map**   The time evolution of Hamiltonian systems is described using a *symplectic map*. To define a symplectic map, we first need to define a symplectic matrix. A matrix $M$ is called symplectic if $M^T A M = A$, where $A = \begin{bmatrix} 0 & \mathbb{I} \\ -\mathbb{I} & 0 \end{bmatrix}$, and $\mathbb{I} \in \mathbb{R}^n$ is the identity matrix. Then, a differentiable map $F(s) : \mathbb{R}^{2n} \to \mathbb{R}^{2n}$ is called symplectic if its Jacobian matrix $J = \frac{\partial F}{\partial s}$

is symplectic everywhere, i.e. $J^T A J = A$. Indeed, that Hamiltonian dynamics are governed by a symplectic map $\frac{d\boldsymbol{s}}{dt} = A \frac{\partial H}{\partial \boldsymbol{s}}$ (see Section 16.3 in [18] for the derivation).

**Canonical transformations**  Physical systems can be equivalently described in any number of arbitrary coordinate systems. For example, the dynamics of a simple mass-spring system can be described equally well using either canonical Cartesian phase space coordinates or the action-angle coordinates (see Fig. 1). A canonical transformation is a mapping that moves between equivalent state spaces in a way that preserves the form of the Hamiltonian dynamics. For example, there is a canonical transformation between the Cartesian phase space coordinates or the action-angle coordinates shown in Fig. 1, since both preserve the correct underlying Hamiltonian dynamics.

More formally, let us consider a Hamiltonian system with a phase-space state $(\boldsymbol{q},\boldsymbol{p})$, a Hamiltonian $\mathcal{H}(\boldsymbol{q},\boldsymbol{p})$ and equations of motion given by Eq. 1. We can transform the state of this system according to new variables[2] $\boldsymbol{Q} = \boldsymbol{Q}(\boldsymbol{q},\boldsymbol{p})$ and $\boldsymbol{P} = \boldsymbol{P}(\boldsymbol{q},\boldsymbol{p})$. In general, such a transformation cannot be expected to preserve the equations of motion in the new variables $(\boldsymbol{Q},\boldsymbol{P})$. However, if there exists a new Hamiltonian $\hat{\mathcal{H}}(\boldsymbol{Q},\boldsymbol{P})$ that describes the same dynamics as the ones described by the original Hamiltonian $\mathcal{H}(\boldsymbol{q},\boldsymbol{p})$, where $\dot{\boldsymbol{Q}} = \frac{\partial \hat{\mathcal{H}}}{\partial \boldsymbol{P}}$ and $\dot{\boldsymbol{P}} = -\frac{\partial \hat{\mathcal{H}}}{\partial \boldsymbol{Q}}$, then such a transformation is called *canonical*. Canonical transformations are also symplectic maps [2]. We provide an illustration of a canonical transformation in Fig. 1 and refer to Stewart [46] for more details.

## 3   Measuring the quality of learnt Hamiltonian dynamics

To measure whether a model has learnt the underlying Hamiltonian dynamics faithfully, we propose measuring whether there exists a canonical transformation between the ground truth phase space and a subspace or a submanifold within the latent space discovered by the model. Note that a direct comparison between the inferred state space and the ground truth phase space (e.g. L2 distance) is impossible because physical systems can be equivalently described in any number of arbitrary coordinate systems and there is no reason why the models should converge to the phase space arbitrarily chosen as the "ground truth" when learning from pixels.

As discussed in Sec. 2, symplectic maps (or canonical transformations) are typically defined over two spaces of the same dimension. In our case, however, the latent space of the model is most often overparametrised to have more dimensions than the ground truth state. Saying this, if the model has learnt to capture the underlying Hamiltonian dynamics, the dynamics in its latent space should "mimic" the dynamics in the ground truth state space, in the following sense: given a latent space trajectory $\boldsymbol{S}(t) \in \mathbb{R}^{2m}$ and a ground state trajectory $\boldsymbol{s}(t) \in \mathbb{R}^{2n}$ where $m \geq n$ there should exist a map $F : \mathbb{R}^{2m} \to \mathbb{R}^{2n}$ such that the energy landscape defined by the learnt Hamiltonian $\mathcal{H}_m(\boldsymbol{S})$ is a scaled version of the energy landscape defined by the original Hamiltonian $\mathcal{H}_n(F(\boldsymbol{S}))$, i.e. $E_m = \mathcal{H}_m(\boldsymbol{S}) = c\mathcal{H}_n(F(\boldsymbol{S})) = cE_n$ (where $c$ is a constant, and $E$ is the energy of the system, see Sec. A.5 in the Appendix for more motivation for the constant). One can think of the map $F$ as the pseudo-inverse of a symplectic map between the space $\mathbb{R}^{2n}$ of the ground truth state and another space $\mathbb{R}^{2n}$ which is embedded in the higher dimensional space $\mathbb{R}^{2m}$ of the model latent space. This pseudo-inverse preserves the Hamiltonian dynamics of the ground truth state space in the model latent space. If we expand the time derivative of the projection of the learnt state space back to the ground truth state space $\hat{\boldsymbol{s}}(t) = F(\boldsymbol{S}(t))$, we see that

$$
\begin{aligned}
\frac{\partial \hat{\boldsymbol{s}}(t)}{\partial t} &= \frac{\partial F}{\partial \boldsymbol{S}} \frac{\partial \boldsymbol{S}(t)}{\partial t} = \frac{\partial F}{\partial \boldsymbol{S}} A_m \frac{\partial \mathcal{H}_m}{\partial \boldsymbol{S}} \\
&= c \frac{\partial F}{\partial \boldsymbol{S}} A_m \frac{\partial F^T}{\partial \boldsymbol{S}} \frac{\partial \mathcal{H}_n}{\partial \boldsymbol{s}} \\
&= c(J A_m J^T) \frac{\partial \mathcal{H}_n}{\partial \boldsymbol{s}}.
\end{aligned}
\tag{2}
$$

Hence, if we can find a mapping $F$ where $c J A_m J^T = A_n$ holds, then it suggests the dynamics in the latent space do in fact "mimic" the dynamics in the ground truth state space, and hence the model has recovered the underlying Hamiltonian dynamics of the system.

---

[2]These can also depend on time but we are not considering time-dependent transformations here. Hence, we will only consider *restricted canonical transformations* in this paper.

To measure how well the model has learnt to mimic the original dynamics, we first learn the mapping $F : \boldsymbol{S} \mapsto \boldsymbol{s}$ that explains as much variance in the original state space $\boldsymbol{s}$ as possible. We then check whether this mapping is symplectic by measuring how much $JA_m J^T$ deviates from $A_n$ up to a constant scaling factor, where $J = \frac{\partial F}{\partial \boldsymbol{S}}$ is the Jacobian of the learnt mapping, and $A_k = \begin{bmatrix} 0 & \mathbb{I} \\ -\mathbb{I} & 0 \end{bmatrix}$ is a $2k \times 2k$ anti-symmetric matrix. We next discuss each of the two steps in more detail.

**Learning to explain the ground truth dynamics** We first learn the mapping $F$ between the model state space $\boldsymbol{S}$ and the ground truth phase space $\boldsymbol{s}$. A naive implementation would use an **MLP** to approximate $F$, however one has to be very careful not to overfit to the training data. In the low data regimes we found that even small MLPs trained with L1 regularization often overfit by learning an unnecessarily complicated map that ends up not being symplectic even if a symplectic map exists (see Sec. A.3 in the Appendix). Hence, we make sure to use at least 1000x more datapoints for training the MLP than the number of its parameters.

Given that learning the map $F$ relies on the availability of ground truth phase space data, which may be hard to obtain in certain domains, we also propose a more data efficient way of learning $F$ that only requires labeling 100 training trajectories. We do so by going back to the basics of machine learning and avoiding overfitting by controlling the expressivity of the mapping $F$ by training a **Lasso regularised linear regression** over a polynomial expansion over the learnt state space features $\boldsymbol{S}$, similarly to DiPietro et al [14]. We start with the expansion order of 1 and progressively increase it up to an upper bound determined by the hyperparameter $\kappa$, until enough variance in $\boldsymbol{s}$ is explained. In our experiments we use $\kappa = 5$ (see Alg. 1 in Appendix for more details). To limit the exponential computational cost of computing higher order polynomial expansions of larger latent spaces, where possible we first filter out "uninformative" state dimensions defined as those dimensions that have average KL from the prior less than the threshold $KL[q_\phi(S_i)|p(S_i)] < 0.01$ as in Duan et al [16], however this step is not necessary, and we have successfully applied our approach without this step in practice to a 1024-dimensional latent space.

The MLP and polynomial regression (PR) instantiations of learning the mapping $F$ have different trade-offs. As discussed above, MLP requires orders of magnitude more ground truth state data (in our experiments at least 640x) than PR, however because it is exposed to more data, it can be more accurate than PR, especially in terms of estimating how much variance in the ground truth dynamics can be explained well by the latent dynamics. The MLP can be arbitrarily expressive, while PR calculations get exponentially more expensive as the expressivity of $F$ is increased. PR, however, has the benefit of interpretability.

Having learnt $F$ using either the MLP or PR method, we calculate its **goodness of fit**, $R^2$, which measures how much information about the ground truth phase space dynamics can be recovered from the learnt state space according to $R^2 = 1 - \frac{\sum(F(\boldsymbol{S}) - \overline{\boldsymbol{s}})^2}{\sum(\boldsymbol{s} - \overline{\boldsymbol{s}})^2}$, where $\overline{\boldsymbol{s}}$ is the mean of the ground truth state variables, $F(\boldsymbol{S})$ are the ground truth state variables predicted from the model state variables $\boldsymbol{S}$, and the summation is over the available data points [13]. $R^2$ serves as a useful estimate of how much information about the ground truth dynamics is preserved in the latent trajectories. We use it instead of estimation error based alternatives (e.g. MSE, which worked equally well in practice), because it is not sensitive to the magnitude of the data, and hence allows for a more direct comparison between scores obtained from different models trained on different datasets.

**Measuring symplecticity of the mapping** We check the symplecticity of the learnt mapping $F$ by calculating $\hat{A}_n = JA_m J^T$. As the next step we could measure directly whether this product is equal to the anti-symmetric matrix $A_n$, however depending on the model it is possible that the product will instead be equal to $A_n^T$ (we are going to drop the subscript for easier readability where the dimensionality can be inferred from context), which may happen if the Hamiltonian learnt by the model is related to the original Hamiltonian through a negative constant $c$ in Eq. 2 (also see Sec. A.3 in the Appendix). Hence, we instead calculate $\hat{A}\hat{A}^T = JAJ^T JA^T J^T$ and measure how far it deviates from the identity matrix:

$$Sym = \text{MSE}(c\hat{A}\hat{A}^T, \mathbb{I}) \tag{3}$$

where $c$ is a normalisation factor as per Sec. 3 (see Sec. A.5 in the Appendix for more motivation), and $\mathbb{I} \in \mathbb{R}^{2n}$. Note that $c$ will be different between different models, since it depends on the learnt

| Dataset | HGN++ | | | | | HGN | | | | |
|---|---|---|---|---|---|---|---|---|---|---|
| | MSE | | VPT | $Sym$ | $R^2$ | MSE | | VPT | $Sym$ | $R^2$ |
| | Rec | Ext | | | | Rec | Ext | | | |
| Mass-spring | **0.05** | **0.18** | **937.0** | **0.0*/0.0*** | **0.99*/1.0*** | 25.07 | 197.67 | 5.75 | 0.68/0.0 | 0.88/0.86 |
| Mass-spring +c | **1.63** | **2.77** | **567.5** | **0.03*/0.04*** | **0.96*/0.95*** | 25.49 | 126.34 | 8.5 | 0.78/0.0 | 0.52/0.5 |
| Pendulum | **1.95** | **10.5** | **85.25** | 1.76/0.61 | 0.54/**0.96** | 25.99 | 294.22 | 1.75 | **0.43**/0.0 | **0.79**/0.81 |
| Pendulum +c | 26.94 | **80.89** | **23.0** | 0.5/0.48 | -0.0/**0.98** | **23.43** | 194.52 | 5.75 | **0.37**/50.0 | **0.39**/0.55 |
| Matching pennies | **2.25** | **11.71** | **640.5** | **0.35**/0.25 | **0.99**/0.98 | 32.46 | 562.98 | 5.75 | 0.55/**0.18** | 0.88/**0.95** |
| Rock-paper-scissors | **4.82** | **34.37** | **146.75** | **0.18**/0.1 | **0.69**/**0.96** | 138.16 | 955.31 | 0.0 | 0.23/0.12 | 0.55/0.88 |
| Double pendulum | 32.24 | **154.83** | **5.5** | **0.28**/0.24 | **0.4**/**0.96** | **26.33** | 196.52 | 2.0 | 0.3/**0.13** | 0.2/0.64 |
| Double pendulum +c | 54.7 | **87.19** | **5.0** | 0.14/0.48 | **0.05**/**0.91** | **26.08** | 97.28 | 2.5 | **0.12**/**0.13** | -0.0/0.6 |
| Two-body | **0.04** | **0.41** | **447.0** | 0.22/0.15 | **0.91**/**0.98** | 22.33 | 152.42 | 12.75 | **0.21**/**0.14** | 0.83/**0.98** |
| Two-body +c | **1.95** | **24.68** | **36.25** | 0.22/0.19 | **0.4**/**0.95** | 25.1 | 69.88 | 15.75 | 0.5/**0.11** | 0.32/0.78 |
| 3D room - circle | 114.43 | **370.05** | 1.0 | **0.11**/max | **0.16**/**0.97** | 39.55 | 1044.6 | **1.25** | **0.11**/max | **0.16**/0.35 |
| MD - 4 particles | 55.68 | 218.47 | 0.75 | **0.07/0.06** | 0.0/**0.94** | 23.41 | 1711.71 | **4.0** | 0.91/0.06 | 0.0/0.82 |
| MD - 16 particles | 364.73 | 447.82 | 0.0 | **0.02/0.02** | 0.0/**0.64** | 5.78 | **396.71** | 3.5 | **0.02**/950.0 | 0.0/0.47 |

Table 1: The reconstruction (Rec) MSE refers to the MSE measured over the first $T = 60$ timesteps, i.e. the same trajectory length that was used for training, but on test data. The extrapolation (Ext) MSE was computed across the subsequent $T$ timesteps, immediately following those used for training. All MSE values are reported as multiples of $10^7$. VPT scores are calculated as the average over forward and backward extrapolation. SyMetric, $Sym$ and $R^2$ results correspond to MLP/PR implementations. * indicates models with SyMetric $= 1$.

Hamiltonian. We estimate it by finding a constant that minimises $Sym$ for each model separately. Since for symplectic maps the right-hand side in Eq. 3 has to be zero everywhere, we need to evaluate it across all avaliable data. In practice we sample $t$ points across $k$ trajectories to evaluate the Jacobian and average $Sym$ score over. The normalisation factor is calculated as $c = 1/\text{mean}(\max(|\hat{A}\hat{A}^T|))$, where the absolute maximum value of $\hat{A}\hat{A}^T$ is taken at every evaluation point, and their average is taken across all the points from a single trajectory. We calculate a different constant for each trajectory to ensure that on datasets where different trajectories are sampled from different Hamiltonians (e.g. mass-spring +c or pendulum +c), $Sym$ is still meaningful. This does not affect $Sym$ for datasets generated from a single Hamiltonian, where in practice the constant is the same across all trajectories.

### 3.1 Aggregating into a single indicator: SyMetric

$R^2$ and $Sym$ measure two orthogonal properties of the learnt dynamics: 1) whether they capture enough information about the ground truth dynamics; and 2) whether the captured dynamics mimic those of the ground truth faithfully. Indeed, these scores can be at the opposite scales of the range in one model. For example, a model may "cheat" and learn to produce perfect trajectories in pixel space by learning $\boldsymbol{q} = [q, \dot{q}, t]$, $\boldsymbol{p} = [0, 0, 0]$, and $\mathcal{H} = p_2$. In this case Hamiltonian dynamics would increase $t$ and, given a powerful enough decoder that can memorise observations indexed by $t$, such a model would score perfectly on any reconstruction based metric, including VPT. Likewise, given a powerful enough function approximator for $F$, this model can also have a perfect $R^2$ score. However, in this case $F$ would not be symplectic, and hence $Sym$ would be high and serve as the only indicator that the model has not actually recovered the underlying Hamiltonian dynamics as hoped. Alternatively, a degenerate $F$ (e.g. the identity function) would be symplectic and result in the perfect $Sym = 0$ score, however in this case if the dynamics learnt by the model do not actually mimic the ground truth dynamics, it will be captured by the low $R^2$. In other words, if $R^2$ is low, then the learnt latent space is degenerate and does not preserve all the information about the ground truth dynamics. In this case, the value of $Sym$ is irrelevant and the model has failed to learn well. If $R^2$ is high but $Sym$ is low, then the learnt dynamics are expressive enough to mimic those of the ground truth, but they are likely to have overfit to the training trajectories and will fail when it comes to extrapolation, since they do not capture the true underlying dynamics. Hence, in order for a model to capture the underlying Hamiltonian dynamics well, the mapping between the latent space and the ground truth phase space needs to be both informative (high $R^2$) and symplectic (low $Sym$).

While our general recommendation is for the practitioners to use the $R^2$ and $Sym$ scores for comprehensive model evaluation, we also propose a way to aggregate the two measures into a single "at-a-glance" score for capturing the main insight: whether a model has discovered the true dynamics or not. We call

this aggregate score the Symplecticity Metric[3] or SyMetric. We define SyMetric to be binary according to

$$\text{SyMetric} = \begin{cases} 1, & \text{if } R^2 > \alpha \wedge Sym < \epsilon \\ 0, & \text{otherwise.} \end{cases} \tag{4}$$

We found that SyMetric parameters $\alpha = 0.9$ and $\epsilon = 0.05$ worked well for both MLP and PR implementations.

## 4 Models

**HGN**   HGN [49] is a generative model that aims to learn Hamiltonian dynamics from pixel observations $\boldsymbol{x}$. It consists of the encoder network that learns to embed a sequence of images $(\boldsymbol{x}_0,...\boldsymbol{x}_T)$ to a lower-dimensional abstract phase space $\boldsymbol{s} \sim q_\phi(\cdot|\boldsymbol{x}_0,...\boldsymbol{x}_T)$ which consists of abstract positions and corresponding momenta $\boldsymbol{s} = (\boldsymbol{q},\boldsymbol{p})$; the Hamiltonian network that maps the inferred phase space to a scalar corresponding to the energy of the system $\mathcal{H}_\gamma(\boldsymbol{s}_t) \in \mathbb{R}$; an integrator that takes in the current state $\boldsymbol{s}_t$, its time derivative calculated using the Hamiltonian network, and the timestep $\Delta t$ to produce the state $\boldsymbol{s}_{t+\Delta t}$; and the decoder network that maps the position coordinates of the phase space back to the image space $p_\theta(\boldsymbol{x}_t) = d_\theta(\boldsymbol{q}_t)$. HGN is trained using the variational objective

$$\frac{1}{T+1}\sum_{t=0}^{T} \mathbb{E}_{p(\boldsymbol{x})}\big[\mathbb{E}_{q_\phi(\boldsymbol{s}_t|X)}[\log p_\theta(\boldsymbol{x}_t|\boldsymbol{s}_t)] - \mathcal{D}_{KL}(q_\phi(\boldsymbol{s}_t|X)|p(\boldsymbol{s}))\big], \tag{5}$$

where $X = \{\boldsymbol{x}_0,...\boldsymbol{x}_T\}$ and $p(\boldsymbol{s})$ is the isotropic unit Gaussian prior.

**HGN++**   In order to improve the performance of HGN, we ran a sweep over hyperparameters, investigated architectural changes (see Sec. A.1 in Appendix for more details) and used our proposed new measures described in the Sec. 3 for model selection. We evaluated a range of learning rates, activation functions, kernel sizes for the convolutional layers, hyperparameter settings for the GECO solution for optimising the variational objective [42] and the inference and reconstruction protocols used for training the model. We also considered larger architectural changes, like whether to use the spatial broadcast decoder [52]; whether to use separate networks for inferring the position and momenta coordinates in the encoder; whether to explicitly encourage the rolled out state at time $t+N$ to be similar to the inferred state at time $t+N$ as in [3]; whether to infer the phase space directly or project it through another neural network as in the original HGN work; whether to train the model to predict rollouts forward in time, or both forward and backward in time as in [24]; and finally whether to use a 2D (convolutional) or a 1D (vector) phase-space and corresponding Hamiltonian. We found that a combination of $3 \times 3$ kernel sizes and leaky ReLU [33] activations in the encoder and decoder, Swish activations [39] in the Hamiltonian network, as well as a 1D phase-space inferred directly from images used in combination with a Hamiltonian parametrised by an MLP network significantly helped to improve the performance of the model. Furthermore, changing the GECO-based training objective to a $\beta$-VAE-based [22] one, training the network for prediction rather than reconstruction and training it explicitly to produce rollouts both forward and backward in time further improved its performance.

## 5 Datasets

We compare the performance of models on 13 datasets instantiating different types of Hamiltonian dynamics introduced in Botev et al [7], including those of mass-spring, pendulum, double-pendulum and two-body toy physics systems, molecular dynamics, dynamics of camera movements in a 3D environment and learning dynamics in two-player non-transitive zero-sum games known to exhibit Hamiltonian-like cyclic behaviour [5, 6, 53]. All of these (apart from 3D room) were instantiated in two different versions of increasing difficulty. See Sec. A.2 in Appendix for more details.

---

[3]While we abuse the notation by calling our proposed performance measure a "metric", we could not resist the pun.

## 6 Baseline measures

A common way to evaluate performance of models of this class is by comparing the mean squared error (MSE) (or equivalent) between the reconstructed and the original pixel observations on a test subset of the data. This has two problems. First, it is well known that pixel reconstruction errors can often be misleading [44, 30, 51, 15] – e.g. low reconstruction errors may be obtained from a perfect reconstruction of the static background while failing to reconstruct the dynamics of interest, if these belong to a relatively small object. Furthermore, each absolute value of the reconstruction error may mean different things across datasets depending on the visual complexity and other particularities of the data, which makes model comparison across datasets hard. Furthermore, good predictive performance on short sequences used for training does not necessarily mean that the model has faithfully captured the underlying dynamics or imply good performance on long time horizon predictions.

**Reconstruction and extrapolation errors**   To demonstrate the problems that arise when using observation-level reconstruction quality for evaluating the model's ability to learn dynamics, we calculate the "reconstruction" pixel MSE – the most commonly used measure of model performance (e.g. employed by [49, 9, 45, 3, 58]), where the model is evaluated on how well it can reproduce the same trajectory length $T$ as was used for training, albeit using test data. We also calculate MSE over extrapolated trajectories, where we continue to roll out the model for a total of $2T$ steps and measure MSE over the last $T$ timesteps, to check whether measuring extrapolation even over short time periods might predict the model's ability to extrapolate further in time better than the "reconstruction" MSE. In all our experiments $T = 60$, and for more fair comparison across datasets in all cases MSE is normalised by the average intensity of the ground truth observation as proposed in Zhong et al [57]: $\text{MSE} = ||\boldsymbol{x} - \hat{\boldsymbol{x}}||_2^2 / ||\boldsymbol{x}||_2^2$, where $\boldsymbol{x}_t$ is the ground truth and $\hat{\boldsymbol{x}}_t$ is the reconstructed observation.

**Valid Prediction Time**   We validate our proposed measures by comparing their predictions to another estimate of the model's performance, namely the *Valid Prediction Time* (VPT) (as in Jin et al [26]). VPT is motivated by the intuition that those models that have truly captured the underlying dynamics of energy conserving non-chaotic systems should in principle be able to produce forward and backward rollouts over significantly longer time intervals than those used for training without significant detriment to the reconstruction quality. Note, however, that VPT may also produce false positive or false negative results as will be discussed in Secs. 3 and 7, and hence is not perfect. It is, however, the best alternative measure we could come up with for validating our proposed measures. To calculate VPT we generate a small number of very long trajectories of between 256 and 1000 steps for our datasets (we use 60 step trajectories for training) and measure how long the model's trajectory remains close to the ground truth trajectory in the observation space. It corresponds to the first time step at which the model reconstruction significantly diverges from the ground truth:

$$\text{VPT} = \underset{t}{\operatorname{argmin}} \left[ \text{MSE}(\boldsymbol{x}_t, \hat{\boldsymbol{x}}_t) > \lambda \right]$$

where $\boldsymbol{x}_t$ is the ground truth, $\hat{\boldsymbol{x}}_t$ is the reconstructed observation at time $t$, and $\lambda$ is a threshold parameter. For our datasets we found a threshold of $\lambda = 0.025$ a reasonable choice based on visual inspection of the rollouts. VPT scores are averaged across 10 trajectories.

## 7 Results

We first trained different variations of the HGN model as described in Sec. 4 on the mass-spring dataset and found that a particular set of hyperparameters and architectural choices consistently resulted in models that were identified by SyMetric as having learnt the underlying Hamiltonian dynamics faithfully. We refer to this improved version of HGN as HGN++. We then trained this model, as well as the original version of HGN, on all 13 datasets. Table 1 demonstrates that HGN++ is indeed a significant improvement on HGN. While HGN is practically unable to produce good rollout trajectories beyond the 60 timesteps that it was trained on, HGN++ can extrapolate well for at least 267 steps forward and 178 steps backward on average across all datasets (vs 11 and 0 steps by HGN) as evidenced by the VPT scores. Backward extrapolation is achieved by feeding $-\Delta t$ to the integrator at test time. Furthermore, on both the easy and hard (+c) versions of the mass-spring dataset, HGN++ is able to faithfully recover the underlying Hamiltonian dynamics, as identified by the SyMetric score of 1.

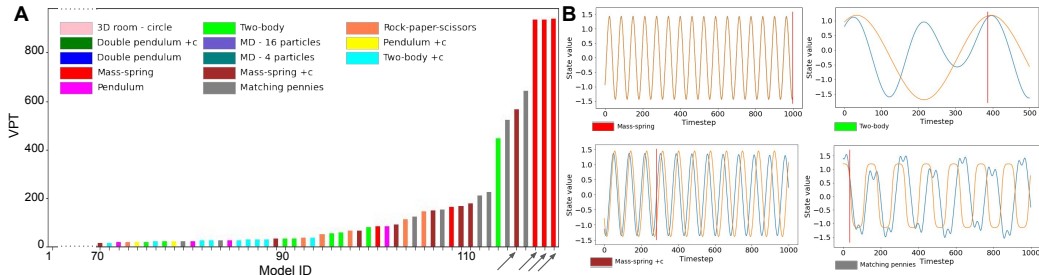

Figure 2: **A**: 119 HGN models trained with different hyperparameters on 13 datasets ordered by average forward and backward VPT score. Arrows point at the models that achieved SyMetric $= 1$. **B**: Example trajectories of a single phase space coordinate in the ground truth phase space (orange) and the projection of the model latent space into the ground truth phase space using $F$ (blue). The models highlighted by SyMetric to have discovered the true Hamiltonian dynamics (trained on mass-spring and mass-spring +c datasets) appear to mimic the ground truth dynamics well, while the other models have qualitatively different dynamics despite sometimes scoring similarly by VPT. Red vertical lines indicate the time step where VPT indicates divergence in pixel space.

Given the radically different learning ability of HGN and HGN++, SyMetric is not the only performance measure that differentiates between them. Indeed, MSE results are also indicative of the superior performance of HGN++. However, not all seeds of HGN++ always converge to the same level of performance, and indeed slight modifications in hyperparameters often affect the model's learning ability in ways that have a negligible effect on MSE, yet can drastically affect whether the models are able to uncover the true Hamiltonian dynamics or not. When we look at the top 20% of model seeds in terms of their VPT performance across all datasets, and compare the different measures in terms of how well they differentiate between the better and worse performing seeds based on how well they correlate with VPT (averaged forward and backward), reconstruction MSE has 0.07 absolute Spearman correlation, extrapolation MSE is better at 0.33, however $R^2$ has stronger correlation of 0.59(0.41) and $Sym$ has correlation of 0.37(0.45) MLP(PR). Hence, when it comes to doing more precise model or hyperparameter selection and differentiating between models that are not obviously failing, it appears that the currently widely used reconstruction MSE measure is very poor, and using either $R^2$ or $Sym$ is more informative.

Fig. 2A orders the 119 models in terms of their average VPT scores. We see that the mass-spring dataset is clearly solved, with a number of HGN++ seeds with slightly different hyperparameters achieving close to perfect 1000-step extrapolations both forward and backward in time after being trained on just 60 steps. The figure also demonstrates that all of these models are picked out by SyMetric as having discovered the true Hamiltonian of the system. However, at the next step change down in VPT measures, we see inconsistent predictions from VPT and SyMetric. A number of model seeds trained on the matching pennies and two-body datasets are scored highly by VPT, but are not highlighted by SyMetric to have discovered the true Hamiltonian, while a HGN++ seed trained on the mass-spring +c data with similar VPT is highlighted as having done so. Which measure is correct: VPT or SyMetric? When we visualise an example trajectory from these models, mapping the learnt latent space to the ground truth through $F$ and visualising both (see Fig. 2B), we see that SyMetric is in fact correct. The latent dynamics of models trained on matching pennies and two-body datasets are qualitatively different from those exhibited by the ground truth system, while the dynamics from the model trained on mass-spring +c appear to be mimicking those of the ground truth.

In order to further verify the validity of SyMetric, Fig. 3A-B visualises four trajectories in the ground truth phase space and in the latent space learnt by the HGN++ seed highlighted as have learnt the underlying Hamiltonian dynamics on the mass-spring +c dataset. It is clear that latent $Q_3$ and its corresponding $P_3$ were able to capture the ground truth state dynamics well. Indeed, when we visualise the extrapolated rollouts of the four sampled trajectories Fig. 3C, we see that they still look good even after 1000 steps, thus verifying that the model is indeed able to produce good rollouts and has appeared to have recovered the underlying ground truth system well. So what causes the relatively low VPT scores recorded for this model? It appears that this discrepancy arises due to a slight error in the estimation of the initial conditions by the model. When we visualise the 1000-step rollout in the ground truth state against the projection of the model's latent state into the ground truth phase space, it is clear

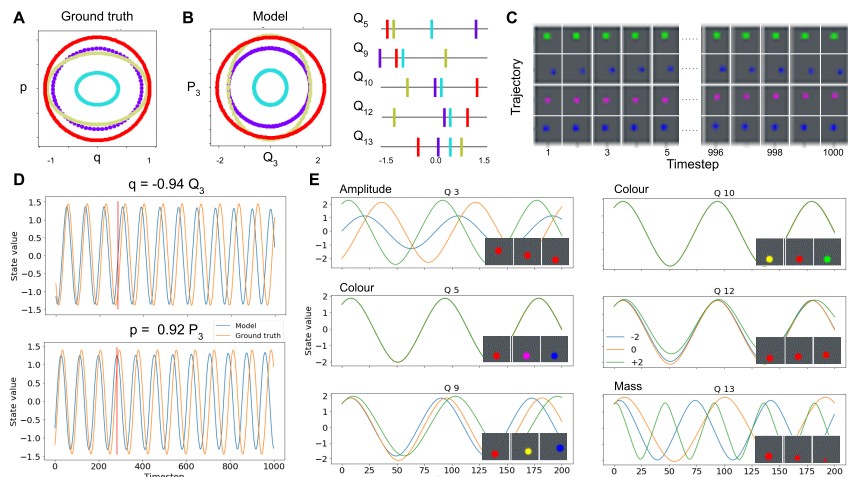

Figure 3: **A**: Visualisation of four trajectories of the mass-spring +c dataset in the ground truth phase space. **B**: Same trajectories visualised in the informative latents of a trained HGN++ with the perfect SyMetric score. If both the $Q_i$ and its corresponding $P_i$ are informative, the trajectories are plotted as a scatter plot. Otherwise they are plotted as points in the corresponding latent dimension. **C**: 1000 step rollouts produced by the model in B of the four trajectories in A-B. The reconstructions are of good quality even after 1000 steps. **D**: Same 1000 step rollouts as in C but shown in the ground truth phase space and the projection of HGN++ from B into the ground truth phase space as produced by the regression step of SyMetric. Due to slight errors in the inference of the initial state, the trajectories diverge after about 300 steps, resulting in a suboptimal VPT score. **E**: Latent traversals of the informative dimensions of HGN++ shown in B. Each plot shows a 200 step rollout trajectory of $Q_3$ when setting the value of the traversed latent dimension to -2, 0 or 2. The inset images demonstrate the corresponding effect on the reconstruction of the first frame of the trajectory (from left to right). Latents $Q_3$, $Q_5$, $Q_{10}$ and $Q_{13}$ appear to have an interpretable meaning, which is indicated in the left top corner of the corresponding subplot.

that the two trajectories are slightly out of phase with each other, which results in a divergence around the time step captured by the VPT score (indicated by the red vertical line in Fig. 3D). Hence, although the model was able to capture the underlying Hamiltonian faithfully as captured by SyMetric score, it appears that its VPT scores were handicapped by the slight errors in the inferred initial conditions.

Furthermore, the HGN++ shown in Fig. 3 had seven informative dimensions at the end of training. Two of them – $Q_3$ and $P_3$ – ended up learning the dynamics that mimic the ground truth phase space. The other 5 latents did not have an informative momentum component, indicating that the model used these latents as constants. When we investigated what these latents learnt to represent by traversing their values and visualising the resulting effect on the dynamics in $Q_3$, we notice that many of these dimensions are physically meaningful – $Q_5$ and $Q_{10}$ learnt to represent variations in colour, while $Q_{13}$ learnt to represent the mass parameter of the Hamiltonian (see relevant equation in Appendix Tbl. 2). Hence, it appears that HGN++ was able to address another criticism of the original HGN [9], i.e. lack of interpretability of its learnt latent space.

# 8  Conclusions

In this paper we have introduced a set of novel performance measures – $R^2$, $Sym$ and their binary combination, SyMetric, which serve as better proxies than the existing measures of whether a model with a Hamiltonian prior has captured the underlying dynamics of a Hamiltonian system from high-dimensional observations, such as pixel images. We have validated our measures both quantitatively – by comparing them to the VPT score, another proxy of the models performance which measures the ability of the model to produce faithful trajectories in extreme extrapolation regimes in the high-dimensional observation space; and qualitatively – by visualising the latent space of the model. We have shown that we were able to use SyMetric to identify a set of hyperparameters and architectural modifications that improved the previously published HGN model in terms of extrapolation performance and interpretability of its latent space, in effect "solving" 2/13 datasets considered here. We

hope that SyMetric can be used by the community to make further progress towards building models with Hamiltonian priors that can learn dynamics from pixels.

While SyMetric requires ground truth phase space information in addition to pixel observations, this information is only required for a relatively small number of trajectories (we used 100 in the PR case), and for relatively short trajectory lengths (60 steps, as was used for training) – the amount of data that would not be sufficient to train the models. This is in contrast to the significantly longer trajectories (256-1000 steps) that were necessary to compute VPT scores – the only observation-space measure to achieve comparable results. By avoiding the need to generate long trajectories, we hope that SyMetric can be a good option for evaluating models that learn from observation where collecting such long trajectories is expensive, e.g. due to the computational costs of running a simulation or other costs corresponding to collecting experimental data.

SyMetric has the drawback of relying on multiple measures: $R^2$ and $Sym$. As part of future work we hope to replace this with a single step by using function approximators for $F$ that are symplectic by design (e.g. 26, 40), thus producing a single measure equivalent to $R^2$, which will only be high if there exists a symplectic mapping between a subspace of the latent space of the model and the ground truth phase space. Furthermore, we hope to extend SyMetric to models that incorporate the Lagrangian prior in their dynamics (e.g. [45, 3, 58]), which should be possible due to the equivalence of the Hamiltonian and the Lagrangian formalisms, and the ability to map between them through the Legendre transform.

Finally, it would be interesting to investigate in detail how SyMetric compares to VPT on systems that are known to exhibit chaotic dynamics (e.g. the MD datasets). This type of problem poses a significant challenge to our models because tiny perturbations to the initial conditions grow exponentially causing trajectories to diverge within a few hundred training steps (e.g. see [19]). While this poses a fundamental challenge to our training procedure and all evaluations metrics, we believe that the significantly shorter trajectories required for SyMetric may be advantageous.

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
