# A  Appendix

## A.1   HGN++

The list of hyperparameters and architectural changes investigated to improve the performance of HGN is listed below. All models were run on P100 GPUs, with one GPU per learner, and each type of evaluator: for calculating test reconstruction losses, SyMetric with PR, SyMetric with MLP and VPT. We used an internal cluster to run the models, and each model was trained for 500k steps.

**Training scheme**: The original HGN was using a sequence of 30 images $(\boldsymbol{x}_1,...\boldsymbol{x}_{30})$ to infer the latent representation $\boldsymbol{z}_1$. The model would then produce a 30 step rollout and the reconstruction error used for training would be calculated between the original data used for inference and the reconstructed $(\hat{\boldsymbol{x}}_1,...\hat{\boldsymbol{x}}_{30})$. Instead we investigated whether the model would work better if the original sequence of images $(\boldsymbol{x}_1,...\boldsymbol{x}_T)$ was used to infer the state at time $\boldsymbol{z}_T$ rather than $\boldsymbol{z}_1$, so that the rollouts used for training are into the future, and the reconstructed frames are not the ones seen by the encoder model. We found that this latter setup worked better for HGN++.

We further investigated how many steps was required for inference: [5-30]; and how many steps the model should reconstruct: $30, 60, 90$. We found that using 5 inference steps and 60 reconstruction steps worked well for HGN++.

Furthermore, we investigated whether it is important to train the model to be able to explicitly produce good rollouts backwards in time (rather than just forward) as per **(author?)** [24]. We found that the forward/backward straining scheme improved the performance of HGN++.

**Encoder type**: The original HGN was using a single encoder network to predict the final state $(\boldsymbol{q},\boldsymbol{p})$. We also investigated the value of other kinds of encoders: 1) separate networks for encoding $\boldsymbol{q}(\boldsymbol{x}_1,...\boldsymbol{x}_T)$ and $\boldsymbol{p}(\boldsymbol{x}_1,...\boldsymbol{x}_T)$; 2) stacked encoder, where the single images are used to infer the position $\boldsymbol{q}(\boldsymbol{x}_i)$, and the inferred position vectors are stacked to infer the momenta $\boldsymbol{p}(\boldsymbol{q}_1,...\boldsymbol{q}_T)$. We found that the original encoder worked the best for HGN++.

The original HGN used an additional MLP to map between the latent sample inferred by the encoder network $\boldsymbol{z} \sim q_\phi(\cdot|\boldsymbol{x}_1,...\boldsymbol{x}_T)$, and the state $\boldsymbol{s} = (\boldsymbol{q},\boldsymbol{p})$ that was unrolled through the Hamiltonian equations of motion according to $\boldsymbol{s}=\text{MLP}(\boldsymbol{z})$. We found that removing this additional transformation and using the encoder sample directly as state worked better for HGN++.

**Decoder type**: We investigated whether replacing a ResNet based decoder with an MLP or a Spatial Broadcast Decoder [52] might be beneficial for HGN. We found that the original ResNet was the best choice.

**Matching inferred states with rolled out states**: similarly to **(author?)** [3] to investigated whether introducing an extra term to the objective function shown in Eq. 5 used to train HGN helps imrove its performance. The extra term forces the state at time step $i$ produced through integrating the equations of motion $\hat{\boldsymbol{s}}_i$ to match the state inferred by the encoder $\boldsymbol{s}_i \sim q_\phi(\cdot|\boldsymbol{x}_{i-T},...\boldsymbol{x}_i)$. We tried different weighing of this additional terms with respect to the original objective function: $\mathcal{L}_{HGN} + \lambda||\hat{\boldsymbol{s}}_i - \boldsymbol{s}_i||_2^2$, with $\lambda \in \{0.01, 0.1, 1, 10\}$. We also tried enforcing this extra constraint every $N \in \{6, 10, \mathbb{U}(T)\}$ steps, where $\mathbb{U}(T)$ stands for uniform sampling of N for each trajectory between 1 and T. We found that none of the choices made a significant difference to HGN training.

**Loss regulariser**: In the original HGN implementation, the variational objective in Eq. 5 was optimised using the GECO solution [42]. We ran a hyperparameter sweep for $\kappa \in [1e-6, 0.01]$. We also considered another option for optimising the variational objective – $\beta$-VAE [22], with a hyperaparameter sweep over $\beta \in [0, 20]$. We found that $\beta$-VAE objective significantly outperformed the GECO one, and $\beta$ around 0.1 or 1 tended to work best for HGN++.

**Latent space**: In the original HGN implementation, the inferred state was 3D: $\boldsymbol{s} \in \mathbb{R}^{4\times4\times32}$. Here we investigated whether a 1D vector based latent state would perform better, e.g. $\boldsymbol{s} \in \mathbb{R}^{32}$. We tried different options for aggregating the original 3D representation into a 1D one: using an MLP, through spatial average or max operation, or through a linear projection. The latter worked the best for HGN++. We also investigated the size of the latent representation and found that 32 dimensions worked as well as the larger options.

**Hamiltonian network**: We investigated whether replacing the Hamiltonian network from a convolutional architecture to an MLP might be beneficial. We tried 1-4 layer MLPs with hidden layer of size

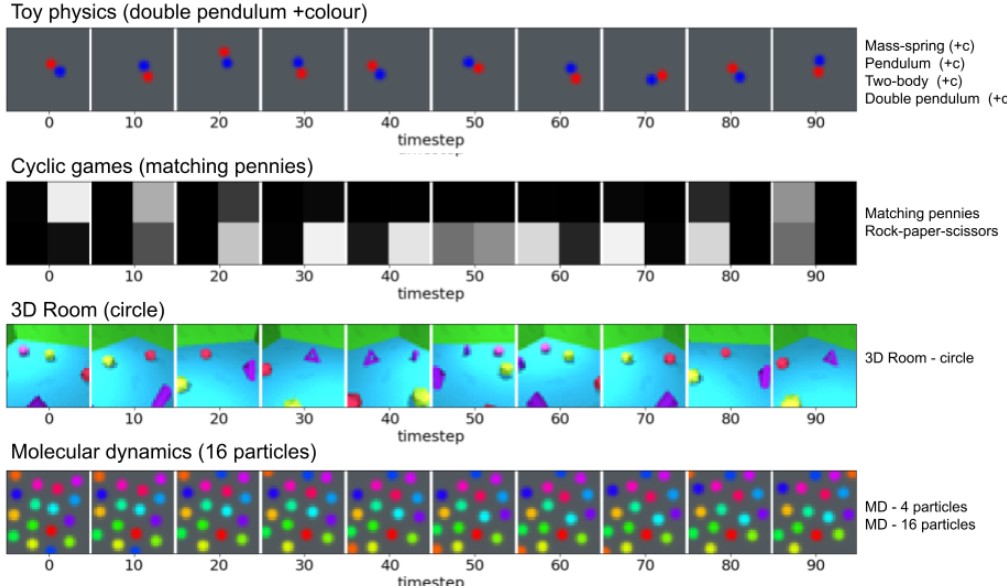

Figure 4: Visualisation of observations from a subset of datasets used in this paper.

250, and found that 4 layers gave us the best performance in HGN++. We also tried adding L1, L2 or a weighted mix of both to the Hamiltonian weights, but found that it did not help. Exchanging Softplus activations with Swish activations on the other hand made a difference.

**Activation function**: We investigated the role of ReLU, leaky ReLU, Softplus and Swish activations in the encoder and decoder of HGN, and found the leaky ReLU worked the best.

**Integrators**: We found taht the original leap-frog integrator worked better than the others considered (Euler, RK4). We found that the original $\Delta t = 0.125$ worked better than other choices (e.g. 0.25).

**Training steps**: The original HGN was trained over 15k steps, We found that increasing the number of steps up to 1mln resulted in better performance. In all reported experiments we used 500k training steps.

**Learning rate**: We found that the original learning rate of 1.5e-4 worked well for our experiments.

**Batch size**: We found that the models were training well with any batch size considered: 10, 32, or 128. The reported experiments used batch size 128.

## A.2  Datasets

All datasets have $32 \times 32 \times 3$ observations. The ground truth phase space is 2D for mass-spring, pendulum and 3D room datasets, 4D for the matching pennies, double pendulum and two-body datasets, 6D for the rock-paper-scissors dataset, 8D for the 4-particle MD dataset, and 32D for the 16-particle MD dataset. Full description of the datasets can be found in Botev et al [**?** ]. A visualisation of example datasets from each type is shown in Fig. 4. All datasets are available under the Apache V2 license here: `https://github.com/deepmind/dm_hamiltonian_dynamics_suite`.

**Toy physics**  We re-use the mass-spring, pendulum and two-body datasets used in the original work on building models with Hamiltonian priors [20, 49], as well as a new double-pendulum dataset. The dynamics in these systems are governed by the Hamiltonians shown in Tbl. 2. The initial conditions for each dataset are sampled in the following way:

- Mass Spring - we sample $q$ and $p$ together from the uniform distribution over the annulus with lower radius bound 0.1 and upper radius bound 1.0 and the we multiply $p$ by $\sqrt{km}$.
- Pendulum - we sample $q$ and $p$ together from the uniform distribution over the annulus with lower radius bound 1.3 and upper radius bound 2.3.
- Double Pendulum - we sample the states of both pendulums analogously to Pendulum.

- Two Body Problem - we follow the same protocol as in [49].

The sampling distribution for every dataset and parameter are shown in Table 3, where $k$,$l$,$m$ and $g$ are the parameters corresponding to the spring force coefficient of the mass-spring system, pendulum length, mass and gravitational force respectively. In the original datasets these values were fixed. We also generated more challenging versions of these datasets, where these parameters are sampled. We also randomly sample the horizontal positions and colours of the masses in both datasets. Although the latter do not affect the resulting dynamics, we wanted to test whether the models are capable of learning in this regime. We denote the more challenging versions of the datasets as "+c". For these datasets the radius of the rendered ball is proportional to its mass (see Fig. 4, top).

| Dataset | Hamiltonian $\mathcal{H}(q,p)$ | Hyperparameters |
|---|---|---|
| Mass Spring | $k\frac{q^2}{2} + \frac{p^2}{2m}$ | $k,m$ |
| Pendulum | $mlg(1-\cos(q)) + \frac{p^2}{2lm}$ | $m,l,g$ |
| Double Pendulum | $\frac{m_2 l_2^2 p_1^2 + (m_1+m_2) l_1^2 p_2^2 - 2m_2 l_1 l_2 p_1 p_2 \cos(q_1-q_2)}{2m_2 l_1^2 l_2^2 (m_1+m_2 \sin(q_1-q_2)^2)} - (m_1+m_2)gl_1\cos(q_1) - m_2 gl_2\cos(q_2)$ | $m_1,m_2,l_1,l_2,g$ |
| N-Body Problem | $-\sum_{i<j} \frac{gm_i m_j}{\|q_i-q_j\|} + \sum_i \frac{\|p_i\|_2^2}{2m_i}$ | $g,m_1,...,m_n$ |

Table 2: The Hamiltonians used for simulating all of the classical mechanics systems.

| Dataset | Hyperparameters |
|---|---|
| Mass Spring | $k=2.0$ 
 $m\sim U(0.2,1.0)$ |
| Pendulum | $m\sim U(0.5,1.5)$ 
 $g\sim U(3.0,4.0)$ 
 $l\sim U(0.5,1.0)$ |
| Double Pendulum | $m\sim U(0.4,0.6)$ 
 $g\sim U(2.5,4.0)$ 
 $l\sim U(0.75,1.0)$ |
| Two Body | $m\sim U(0.5,1.5)$ 
 $h\sim U(0.5,1.5)$ |

Table 3: Sampling protocol for the hyperparameters of the coloured Toy physics datasets.

**Cyclic games**   The Multi-agent cyclic games dataset are generating by using the continuous-time two population replicator dynamics are defined as:

$$\dot{x}_i = x_i\left[(\boldsymbol{Ay})_i - \boldsymbol{x}^T\boldsymbol{Ay}\right]$$
$$\dot{y}_j = y_j\left[(\boldsymbol{x}^T\boldsymbol{B})_j - \boldsymbol{x}^T\boldsymbol{By}\right]$$

(6)

where, $\boldsymbol{A}=-\boldsymbol{B}$ are the payoff matrices of a zero-sum game for the row and column player respectively, and $(\boldsymbol{x}, \boldsymbol{y})$ the joint strategy profile. We generate ground-truth trajectories by integrating the coupled set of ODEs equation 6 using an improved Euler scheme or RK45. In both cases the ground-truth state, i.e., joint strategy profile (joint policy), and its first order time derivative, is recorded at regular time intervals $\Delta t$. Trajectories start from uniformly sampled points on the product of the policy simplexes. No noise is added to the trajectories.

**Molecular Dynamics**   The goal of this set of datasets is to benchmark the performance of models on problems involving complex many-body interactions. In particular, the generated datasets employ a Lennard-Jones (LJ) potential [27], which is a popular benchmark problem and an integral part of more complex force fields used, for example, to model water [1] or proteins [10].The two generated LJ datasets have increasing complexity: one comprising only 4 particles at a very low density and another one for a 16-particle liquid at a higher density. These MD datasets are rendered using the same scheme as the toy physics datasets. All masses are set to unity and particles are represented by circles of equal size with a radius value adjusted to fit the canvas well. In addition, different colours are assigned to the particles to facilitate tracking their trajectories.

**3D Room** To evaluate the ability of models to deal with complex 3D visuals, we use a dataset of MuJoCo [47] scenes consisting of a camera moving around a room with 5 randomly placed objects. The objects are sampled from four shape types: a sphere, a capsule, a cylinder and a box. Each room was different due to the randomly sampled colours of the wall, floor and objects similar to [28]. The dynamics are created by rotating the camera around a single randomly sampled parallel of the unit hemisphere centered around the center of the room. The rendered scenes are used as observations, and the Cartesian coordinates of the camera and its velocities estimated through finite differences are used as the state.

## A.3 SyMetric

### A.3.1 Jacobian calculations

We have a Jacobian

$$
J = \begin{bmatrix} \frac{\partial q}{\partial Q} & \frac{\partial q}{\partial P} \\ \frac{\partial p}{\partial Q} & \frac{\partial p}{\partial P} \end{bmatrix}
$$

and an anti-symmetric matrix

$$
A = \begin{bmatrix} 0 & 1 \\ -1 & 0 \end{bmatrix}
$$

According to Eq. 2 we also have that $cJAJ^T = A$ if $J$ is the Jacobian of a symplectic map.

Writing out the calculation

$$
JAJ^T = \begin{bmatrix} \frac{\partial q}{\partial Q} & \frac{\partial q}{\partial P} \\ \frac{\partial p}{\partial Q} & \frac{\partial p}{\partial P} \end{bmatrix} \begin{bmatrix} 0 & 1 \\ -1 & 0 \end{bmatrix} \begin{bmatrix} \frac{\partial q}{\partial Q} & \frac{\partial p}{\partial Q} \\ \frac{\partial q}{\partial P} & \frac{\partial p}{\partial P} \end{bmatrix} =
$$

$$
= \begin{bmatrix} -\frac{\partial q}{\partial P} & \frac{\partial q}{\partial Q} \\ -\frac{\partial p}{\partial P} & \frac{\partial p}{\partial Q} \end{bmatrix} \begin{bmatrix} \frac{\partial q}{\partial Q} & \frac{\partial p}{\partial Q} \\ \frac{\partial q}{\partial P} & \frac{\partial p}{\partial P} \end{bmatrix} =
$$

$$
= \begin{bmatrix} -\frac{\partial q}{\partial Q}\frac{\partial q}{\partial P} + \frac{\partial q}{\partial Q}\frac{\partial q}{\partial P} & -\frac{\partial q}{\partial P}\frac{\partial p}{\partial Q} + \frac{\partial q}{\partial Q}\frac{\partial p}{\partial P} \\ -\frac{\partial p}{\partial P}\frac{\partial q}{\partial Q} + \frac{\partial p}{\partial Q}\frac{\partial q}{\partial P} & -\frac{\partial p}{\partial P}\frac{\partial p}{\partial Q} + \frac{\partial p}{\partial P}\frac{\partial p}{\partial Q} \end{bmatrix} =
$$

$$
= \begin{bmatrix} 0 & -\frac{\partial q}{\partial P}\frac{\partial p}{\partial Q} + \frac{\partial q}{\partial Q}\frac{\partial p}{\partial P} \\ -\frac{\partial q}{\partial Q}\frac{\partial p}{\partial P} + \frac{\partial q}{\partial P}\frac{\partial p}{\partial Q} & 0 \end{bmatrix} =
$$

$$
= \begin{bmatrix} 0 & -\left(\frac{\partial q}{\partial P}\frac{\partial p}{\partial Q} - \frac{\partial q}{\partial Q}\frac{\partial p}{\partial P}\right) \\ \frac{\partial q}{\partial P}\frac{\partial p}{\partial Q} - \frac{\partial q}{\partial Q}\frac{\partial Pp}{\partial P} & 0 \end{bmatrix} = \hat{A}
$$

If the constant $c$ in Eq. 2 is negative, the above turns into

$$
JA^TJ^T = \begin{bmatrix} \frac{\partial q}{\partial Q} & \frac{\partial q}{\partial P} \\ \frac{\partial p}{\partial Q} & \frac{\partial p}{\partial P} \end{bmatrix} \begin{bmatrix} 0 & -1 \\ 1 & 0 \end{bmatrix} \begin{bmatrix} \frac{\partial q}{\partial Q} & \frac{\partial p}{\partial Q} \\ \frac{\partial q}{\partial P} & \frac{\partial p}{\partial P} \end{bmatrix} =
$$

$$
= \begin{bmatrix} \frac{\partial q}{\partial P} & -\frac{\partial q}{\partial Q} \\ \frac{\partial p}{\partial P} & -\frac{\partial p}{\partial Q} \end{bmatrix} \begin{bmatrix} \frac{\partial q}{\partial Q} & \frac{\partial p}{\partial Q} \\ \frac{\partial q}{\partial P} & \frac{\partial p}{\partial P} \end{bmatrix} =
$$

$$
= \begin{bmatrix} \frac{\partial q}{\partial Q}\frac{\partial q}{\partial P} - \frac{\partial q}{\partial Q}\frac{\partial q}{\partial P} & \frac{\partial q}{\partial P}\frac{\partial p}{\partial Q} - \frac{\partial q}{\partial Q}\frac{\partial p}{\partial P} \\ \frac{\partial p}{\partial P}\frac{\partial q}{\partial Q} - \frac{\partial p}{\partial Q}\frac{\partial q}{\partial P} & \frac{\partial p}{\partial P}\frac{\partial p}{\partial Q} - \frac{\partial p}{\partial P}\frac{\partial p}{\partial q} \end{bmatrix} =
$$

$$
= \begin{bmatrix} 0 & \frac{\partial q}{\partial P}\frac{\partial p}{\partial Q} - \frac{\partial q}{\partial Q}\frac{\partial p}{\partial P} \\ \frac{\partial q}{\partial Q}\frac{\partial p}{\partial P} - \frac{\partial q}{\partial P}\frac{\partial p}{\partial Q} & 0 \end{bmatrix} =
$$

$$
= \begin{bmatrix} 0 & \frac{\partial q}{\partial P}\frac{\partial p}{\partial Q} - \frac{\partial q}{\partial Q}\frac{\partial p}{\partial P} \\ -\left(\frac{\partial q}{\partial P}\frac{\partial p}{\partial Q} - \frac{\partial q}{\partial Q}\frac{\partial p}{\partial P}\right) & 0 \end{bmatrix} = \hat{A}^T
$$

Setting

$$a = \frac{\partial q}{\partial P}\frac{\partial p}{\partial Q} - \frac{\partial q}{\partial Q}\frac{\partial p}{\partial P}$$

we get

$$\hat{A}\hat{A}^T = \begin{bmatrix} (-a)^2 & 0 \\ 0 & a^2 \end{bmatrix} = \begin{bmatrix} a^2 & 0 \\ 0 & a^2 \end{bmatrix}$$

If the original Jacobian $J$ was that of a symplectic map then there should exist a constant $c$ for which $ca^2 = 1$ everywhere.

### A.4 $Sym$ score calculation details

**Polynomial regression**  To make the calculation of SyMetric computationally tractable we limit the maximal polynomial expansion threshold $\kappa = 5$. Furthermore, when evaluating the Jacobian in Eq. 3, we ignore the Jacobian terms with weight below $\epsilon = 1e-3$ to save computation. We also set the threshold for the goodness of fit parameter $R^2$ to $\alpha = 0.9$ to stop polynomial expansion and Lasso regression computation as soon as more than 90% of the variance in the ground truth trajectories is explained. We use the sklearn implementation of Lasso regression with maximum iteration number set to 1000, cross-validation set to 2, and Lasso $\alpha \in \{1e-8, 1e-7, 1e-6, 1e-5, 1e-4, 1e-3, 1e-2\}$. All trajectories are standardized before running the regression.

We find that in general the longer the trajectory on which the metric is calculated, the more accurate the results, however the accuracy on shorter trajectories (e.g. of 60 steps used in this paper) can be increased by calculating the metric on a number of samples and reporting the maximum or rounded average of the obtained score. In all our experiments we calculated the metric over 20 samples of 5 trajectories each. We found no improvements when we increased the number of trajectories within a sample.

When we use MLP instead of Lasso regression to measure SyMetric using the same amount of data, we find that the model overfits and the resulting symplecticity scores are very noisy $Sym = 0.38 \pm 0.18$, when the same model evaluated using the polynomial method gets $Sym = 0.0001 \pm 0.00005$, the latter score also being more indicative of the model's ability to extrapolate.

**MLP**  To implement $F$ as an MLP, we use 4 hidden layers with 4 units each, tanh nonlinearity, and Adam optimizer with $1.5e-3$ learning rate. To ensure that we do not overfit, we calculate the number of parameters in the MLP, and collect 1000x the parameter number of trajectories of 60 steps each. We then standardize the resulting data and split the resulting training set into randomly sampled mini-batches of size 64 for training the MLP, and train it for 10,000 steps. We also use L1 regularization on the weights with $\lambda = 0.01$. For calculating $Sym$, we use 50 training trajectories sampled at 10 random points each. To calculate SyMetric, we use $\alpha = 0.9$ and $\epsilon = 0.05$. See Alg. 2 for more details.

### A.5 Understanding the constant

There are several ways to understand the need for the constant $c$ in Eq. 3. First, since the learnt latent space is not grounded to the "ground truth", it can in effect learn to represent the state using different "units". Then the model will be representing exactly the same system in its latent space, but the energy values for all the trajectories will be orders of magnitude different from those of the ground truth trajectories, resulting in $E_m = cE_n$. Saying this, if all trajectories were related to each other with the same constant, then the mapping $F$ could learn what the constant is, and one would expect $H_m(S) == H_n(F(S))$ to hold. This would be true if we were optimising for MSE when learning $F$, but since we optimise for variance explained ($R^2$), the magnitudes of $F(S)$ and $s$ may be different, as long as they correlate. Second, some of the datasets used in the paper re-sample the constants of the Hamiltonian for each trajectory (e.g. mass in mass-spring +c). What this means is that the model is in effect learning multiple Hamiltonians from the same family. While in theory the model should be able to learn to infer the right constants from the pixel observations, as it does with the energy, so that $c = 1$, we choose not to penalise it for not doing so, and instead learning multiple Hamiltonians with different "units".

**Algorithm 1:** SyMetric algorithm using polynomial regression (PR).

---

**Data:** $K$ ground truth state trajectories of $T$ time steps $s = (q, p) \in \mathbb{R}^{K \times T \times 2n}$, $K$ learnt latent state trajectories of $T$ time steps $S = (Q, P) \in \mathbb{R}^{K \times T \times 2m}$, maximum polynomial expansion order $\kappa$, minimal acceptable goodness of fit $\alpha$, maximum acceptable deviation for symplecticity $\epsilon$

**Result:** SyMetric, Sym, $R^2$

**begin**

    $p_{exp} \longleftarrow 1, ;$

    $R^2 \longleftarrow 0, ;$

    $Sym \longleftarrow \emptyset;$

    $A \in \mathbb{R}^{2n \times 2n} \longleftarrow \begin{bmatrix} 0 & \mathbb{I} \\ -\mathbb{I} & 0 \end{bmatrix};$

    $S \longleftarrow$ `RemoveUninformativeDims(`$S$`)`;

    **while** $p_{exp} < \kappa$ **and** $R^2 < \alpha$ **do**

        $S^* \longleftarrow$ `PolynomialExp(`$S$`, `$p_{exp}$`)`;

        $F \longleftarrow$ `Lasso(`$S^*$`, `$s$`)`;

        $R^2 \longleftarrow$ `VarianceExplained(`$F(S^*)$`, `$s$`)`;

        $p_{exp} \longleftarrow p_{exp} + 1$ ;

    **end**

    $J \longleftarrow \frac{\partial F}{\partial S}$

    **for** $k \in K$ **do**

        $\hat{\mathbb{I}} \longleftarrow \emptyset;$

        **for** $t \in T$ **do**

            $J_{kt} \longleftarrow$ `Evaluate(`$J$`, `$S^*_{kt}$`)`;

            Add $J A J^T (J A^T J^T)^T$ to $\hat{\mathbb{I}}$;

        **end**

        $c \longleftarrow$ `CalculateNormalisingConst(`$\hat{\mathbb{I}}$`)`;

        Add `MSE(`$c\hat{\mathbb{I}}$`, `$\mathbb{I}$`)` to $Sym$;

    **end**

    $Sym \longleftarrow$ `Mean(`$Sym$`)`;

    **if** $R^2 > \alpha$ **and** $Sym < \epsilon$ **then**

        SyMetric $\longleftarrow 1$

    **else**

        SyMetric $\longleftarrow 0$

    **end**

**end**

---

**Algorithm 2:** SyMetric algorithm using MLP with ridge regularization.

---

**Data:** $K$ ground truth state trajectories of $T$ time steps $s = (q,p) \in \mathbb{R}^{K \times T \times 2n}$, $K$ learnt latent state trajectories of $T$ time steps $S = (Q,P) \in \mathbb{R}^{K \times T \times 2m}$, maximum polynomial expansion order $\kappa$, minimal acceptable goodness of fit $\alpha$, maximum acceptable deviation for symplecticity $\epsilon$

**Result:** SyMetric, Sym, R$^2$

**begin**

    $p_{exp} \longleftarrow 1,\,;$

    $R^2 \longleftarrow 0,\,;$

    $Sym \longleftarrow \emptyset;$

    $A \in \mathbb{R}^{2n \times 2n} \longleftarrow \begin{bmatrix} 0 & \mathbb{I} \\ -\mathbb{I} & 0 \end{bmatrix};$

    $S \longleftarrow \texttt{RemoveUninformativeDims}(S);$

    $F \longleftarrow \texttt{MLP}(S^*, s);$

    $J \longleftarrow \frac{\partial F}{\partial S}$

    **for** $k \in K$ **do**

        $\hat{\mathbb{I}} \longleftarrow \emptyset;$

        **for** $t \in T$ **do**

            $J_{kt} \longleftarrow \texttt{Evaluate}(J, S^*_{kt});$

            Add $JAJ^T(JA^TJ^T)^T$ to $\hat{\mathbb{I}};$

        **end**

        $c \longleftarrow \texttt{CalculateNormalisingConst}(\hat{\mathbb{I}});$

        Add $\texttt{MSE}(c\hat{\mathbb{I}}, \mathbb{I})$ to $Sym;$

    **end**

    $Sym \longleftarrow \texttt{Mean}(Sym);$

    **if** $R^2 > \alpha$ **and** $Sym < \epsilon$ **then**

        SyMetric $\longleftarrow 1$

    **else**

        SyMetric $\longleftarrow 0$

    **end**

**end**

---