# OpenReview forum: "SyMetric: Measuring the Quality of Learnt Hamiltonian Dynamics Inferred from Vision"
_NeurIPS.cc/2021/Conference — NeurIPS 2021 Poster_

### Official Review · Reviewer_VPSk · 2021-06-28

**Rating:** 6
**Confidence:** 4

**Summary:**

The paper proposes a distance measure to measure the symplecticness of the Hamiltonian Graph Networks. The experiments show that this measure is rarely equal to 1. However, it is only True for models that generally perform good on the task.

**Main Review:**

I already reviewed the paper for ICML. The paper has been improved and the minor aspects were addressed. However, the main shortcomings have not been addressed (but most likely also cannot be addressed). The main shortcomings are:

* the similarity measure is too complex to compute. It requires an additional function approximator to compute the measure. Furthermore, this fitting is sensible to the choice of the black-box function approximator.


* the similarity measure is binary and only positive for two environments. Therefore, it is very hard to evaluate whether the distance measure is of any use or provides any intuition. It simply just shows that the currently learned latent codes are not that great.

To be honest, I do not think that there is any path to fix these two shortcomings without fundamentally looking for a new approach and evaluation. The approach needs to be improved such that the results increase my confidence in this measure as it is really hard to interpret. Due to these shortcomings, the approach lacks generality and I do not believe that the approach will make much of an impact on the community. However, the execution of the paper is very well done. The paper is well written, has nice figures and the evaluation is done thoroughly. Therefore, one cannot complain about the paper except the significance of the results. My personal take-away is that this paper provides a good documentation of performed research which leads to non-significant results. Therefore, I believe that the paper is valuable because of the good documentation, but its not significant to be presented at NeuRIPS. This conclusion is subjective and hence, please take it with a grain of salt. If the other reviewers think that the paper provides significant results please ignore my review.

### Post Rebuttal Comment:
The comments of the authors clarified many points. I still cannot champion the paper for acceptance. As there is nothing wrong with the paper and its mostly about contribution and impact, I am happy with a weak accept. However, this rating is only marginally above the borderline. Therefore, I will not argue for the acceptance.


**Time Spent Reviewing:**

1

---

> ### Author Response · Authors · 2021-08-10
> **Response**
>
> We thank the reviewer for reviewing our paper again. We are sorry to hear that the reviewer found our updates minor. In the previous version of the paper we only included SyMetric results using polynomial regression (PR), and evaluated on 6 datasets. The new version of the paper now also includes results of SyMetric implemented using an MLP, and discusses the tradeoffs between the MLP and the PR version (e.g. data efficiency). It presents the results evaluated on 13 datasets, analysing the performance of 159 models, and demonstrates that both the binary SyMetric (calculated using PR or MLP) and its continual constituent metrics (R^2 and Sym) are better correlates of model extrapolation performance than the alternative reconstruction-based metrics typically used in the literature.
>
> The reviewer claims that the main two shortcomings of our paper are the following:
>
> 1) The similarity measure is too complex to compute. It requires an additional function approximator to compute the measure. Furthermore, this fitting is sensible to the choice of the black-box function approximator.
> 2) The similarity measure is binary and only positive for two environments. Therefore, it is very hard to evaluate whether the distance measure is of any use or provides any intuition. It simply just shows that the currently learned latent codes are not that great.
>
> To address the first point, we have significantly simplified the similarity measure computation by replacing the PR pipeline with a simple MLP. We are not sure how this step can be simplified any further. Given that our measures work with at least two different choices of black-box function approximator (PR and MLP), both of which are very general, we are not sure why the reviewer believes that our measures are sensitive to the choice of the function approximator. Finally, we are not the first to propose evaluation measures that require function approximation to compute. Many other metrics routinely used in ML also rely on learnt function approximation. For example, many of the metrics for evaluating generative models, like GANs (average log-likelihood score that requires using KDE (Goodfellow et al, 2014); Inception score which relies on a pre-trained classifier (Salimans et al, 2016, Heusel et al, 2017)); many of the disentanglement metrics (like DCI or UDR which require learning a mapping between the learnt latents and the ground truth generative factors (Eastwood and Williams, 2018; Duan et al, 2020); or MIG that relies on estimating the mutual information between the two (Chen et al, 2018)); and even the scoring functions for AlphaFold (Senior et al, 2021). We believe that as a ML community, we are well equipped to detect and deal with any potential model fitting problems, and hence this is not a limitation of the approach.
>
> While in the original version of the paper we concentrated on the binary version of the measure, in the updated version we analysed the two scalar constituent scores (R^2 and Sym) in more detail, and demonstrated that they are very informative of model performance. More so than the existing MSE based measures (see lines 286-294): ​​ “reconstruction MSE has 0.07 absolute Spearman correlation, extrapolation MSE is better at 0.33, however R^2 has stronger correlation of 0.59(0.41) and Sym has correlation of 0.37(0.45) MLP(PR).” In the new version of the paper we also describe the importance of the scalar metrics in helping users understand in more detail how well the models are doing.  E.g. if R^2 is low, then their learnt latent space has failed to capture all of the structure of the ground truth, and the value of Sym is irrelevant. If R^2 is high and Sym is low, then the learnt dynamics are expressive enough to mimic those of the ground truth, but they won’t extrapolate well, as the captured dynamics are different from the ground truth dynamics. Only if the latent dynamics are both informative of the ground truth latent dynamics (high R^2) and the mapping between the two is symplectic (high Sym) has the model captured the true Hamiltonian. The binary combination of the two (with their corresponding thresholds) is only suggested as a way to also give users a shorthand for an "at a glance" kind of understanding of how well their models have learnt. It is not meant to be prescriptive.
>
> We really appreciate the reviewer’s compliments on the paper presentation, however we would like to understand what kinds of results would increase the reviewer’s confidence in our proposed measures. We have already demonstrated that they are useful for model selection when presented with 159 models trained on 13 datasets, and that they are qualitatively better and model detection than all other pixel-based measures, even VPT, that requires almost 20x longer trajectories to compute.
>
> We believe that our demonstration that the currently learned latent codes are not that great is also an important contribution of this work, as this was not clear from prior work. Previous work, which only looked at the reconstruction quality over the same trajectory length as was used for training the model, created the impression that these models learn the underlying dynamics well. However, our work demonstrates that this is not the case and that as a community we have a long way to go in terms of improving the class of models that attempt to learn dynamics using priors from classical mechanics from high-dimensional observations.
>
> We believe that our paper provides significant results because as it stands the community believes that the existing models are good at learning Hamiltonian dynamics from pixels. Our measure is the first one to systematically identify that this story is misleading. Furthermore, as a field we currently have no good way to measure the quality of the learnt dynamics in models that learn them from pixels, which limits progress and precludes these models from being used in practical applications, like robotics or self-driving cars. While our proposed measure may not be perfect primarily due to relying on a small amount of ground truth state data, it is the first and currently the only way to measure the quality of the learnt dynamics and hence can serve as a stepping stone for more research developing other measures in the future.

---

> > ### Comment · Reviewer_VPSk · 2021-08-31
> > **Reviewer Response**
> >
> > Thanks for the extensive reply. I greatly appreciate that you put so much more effort than many other authors into the re-submission. I think you did a good job with this paper but I also cannot champion the paper. As noted by the other reviewers and me, this measure has too many moving parts that need to be aligned to work (which IMO is a bad for a measure).
> >
> > If this would be a Journal, I would be happy to accept the work as there a no measure flaws and the research is well executed. For a space constrained conference venue I am exactly borderline. If there is sufficient space, the paper should be accepted. If there are too many too good papers, I would not complain that this paper is rejected.

---

> > > ### Author Response · Authors · 2021-09-01
> > > **Thank you**
> > >
> > > Dear Reviewer,
> > >
> > > Thank you for acknowledging our response and increasing your score. We really appreciate it.

---

### Official Review · Reviewer_xPfe · 2021-07-11

**Rating:** 6
**Confidence:** 4

**Summary:**

This paper mainly focuses on the application of Hamiltonian Dynamic in computer vision tasks. The authors propose a new metric to evaluate how the learned latent space coincide with the ground truth. Based on the novel metric, they further improve the previous HGN with multiple modifications in different aspects, leading to much better performance.

**Limitations And Societal Impact:**

1.	The main contribution of this paper is the new evaluation metric SyMetric. However, two things which I think are important for this metric are not explained in the paper.
a)	The final design of SyMetric is a binary indicator function. In this way, how to rank multiple models when only SyMetric is provided without $R^2$ and Sym?
b)	Although the authors conduct extensive experiments on a bunch of dataset, all results are based on the method of HGN and other variants. Is there any chance that the proposed metric cannot generalize to other type of methods? I suggest extra experiments on works like [8, 44, 2, 57] to further prove this.
2.	Maybe it is better to place the introduction of HGN++ after the new metric is explained for better understanding.
3.	The authors mention that the design of HGN++ is selected using ‘proposed new measures’ (L106-107). However, it is not clear that:
a)	Which measurement is adopted during selection, Sym, $R^2$ or SyMetric? As I can understand, the HGN++ is designed so that one of these three metrics is consistently better. As explained in this paper, Sym and $R^2$ must be used together for comprehensive evaluation. However the proposed SyMetric is binary as mentioned above, thus unable to select among multiple choices. Moreover, the Sym of HGN++ is not consistently better than HGN among all datasets as shown in Tab. 1 which means the model selection may not be conducted based on Sym.
b)	What strategy is used in the model selection? How to find the best combination of the different modification?
4.	Is it safe enough to get the claim ‘SyMetric is in fact correct’ (L305) based on one sample visualization in Fig. 2B?


**Main Review:**

1.	The authors provide a comprehensive analysis on the drawbacks of MSE as measurement for the corresponding task.
2.	Rather than considering image-level metric, the authors take into account the canonical transformation for Hamiltonian dynamics, which is interesting and novel.
3.	The proposed method is proven to be useful when considering HGN as baseline.


**Time Spent Reviewing:**

15 hours

---

> ### Author Response · Authors · 2021-08-10
> **Response**
>
> We thank the reviewer for their  helpful questions and comments.
>
> - "The final design of SyMetric is a binary indicator function. In this way, how to rank multiple models when only SyMetric is provided without R2 and Sym?"
>
> Actually we do not prescribe SyMetric as a binary measure without R^2 and Sym. This is also why we did not include SyMetric in Table 1. Instead we present both R^2 and Sym together. We believe that only both metrics taken together provide a comprehensive picture of how well different models perform.  If R^2 is low, then the learnt latent space is losing information compared to the ground truth, and the value of Sym is irrelevant. If R^2 is high and Sym is low, then the learnt dynamics are expressive enough to mimic those of the ground truth, but they won’t extrapolate well, as the captured dynamics are different from the ground truth dynamics. Only if the latent dynamics are both informative of the ground truth latent dynamics (high R^2) and the mapping between the two is symplectic (high Sym) has the model captured the true Hamiltonian dynamics well. Hence, we suggest that the practitioners use both of these scalars to gain a detailed understanding of how well their models are doing. The suggested binary combination of the two (with their corresponding thresholds) is our way to also give users a shorthand for an "at a glance" kind of understanding of how well their models have learnt.
>
> - "Is there any chance that the proposed metric cannot generalize to other type of methods? I suggest extra experiments on works like [8, 44, 2, 57] to further prove this."
>
> Since the method does not make any assumptions on the model class,  apart from the fact that it will be learning Hamiltonian dynamics, and since it only requires the trajectories from the learnt latent space as input, we are confident that it will work for any method that learns dynamics with the Hamiltonian prior. We would have happily applied it to other model classes, however [8] is the same as HGN (the only difference is that it doesn't use stochastic latents, and we tried this as part of our HGN++ hyperparameter sweep), while [4], [2] and [57] use the Lagrangian formalism in their models. While our approach should generalise to the Lagrangian formalism, it will require a modification to take into the account the Legendre transform to convert the learnt state space in the Lagrangian formalism (q, q_dot) to the phase space used in the Hamiltonian formalism (q, p). We were running out of space as it was, so left this step to future work.
>
> - "a) Which measurement is adopted during selection, Sym, R2 or SyMetric? b) What strategy is used in the model selection? How to find the best combination of the different modification?"
>
> We used the binary SyMetric for this particular task. Note that Sym on its own is not informative. If R^2 is low, it means that the mapping F is degenerate, so its symplecticity is not meaningful.
>
> - "Is it safe enough to get the claim ‘SyMetric is in fact correct’ (L305) based on one sample visualization in Fig. 2B?"
>
> We have of course done a much more in depth qualitative analysis of that model than just a visualisation of a single trajectory as presented in Fig 2B, but due to the limit of space it was not possible to include anything else in the main text. Apart from looking at more trajectories in latent space, we also visualised up to 10k step extrapolations in pixel space (note that the model only saw 60 steps during training), and performed an in depth analysis of the learnt latent space as shown in Fig 3. All of those analyses made us confident that the model did in fact learn the underlying Hamiltonian correctly. We are happy to include more in depth analysis in the supplementary materials in the updated version of the paper.
>
> - "Maybe it is better to place the introduction of HGN++ after the new metric is explained for better understanding."
>
> We will do that in the updated version of the paper.

---

### Official Review · Reviewer_sUUp · 2021-07-17

**Rating:** 6
**Confidence:** 4

**Summary:**

The paper suggests novel measures for evaluating latent dynamics model by measuring how close the latent dynamics is to being governed by a symplectic map, a property of Hamiltonian dynamics. The paper analyses variants of the original Hamiltonian Generative Network [48] using the model and identifies better variants. The measure is also compared to related measures such as the MSE image reconstruction error, or the valid prediction time (VPT).


**Ethical Concerns:**

none apparent

**Limitations And Societal Impact:**

The authors mention several limitations of their work in the discussion section. Further discussion is needed according to the limitations discussed in sec "weaknesses" above.

**Main Review:**


Strengths:
- The proposed SyMetric measure is novel and can be interesting for theoretical analysis of learning latent dynamics in simulation environments.
- The identified improvements over HGN are a secondary contribution.

Weaknesses:
- Sec 2, l74 ff is not well comprehensible. A differentiable map F is proposed whose symplectic property is defined. No relation to the Hamiltonian is made. The last sentence in this subsection in l. 78/79 uses the term "symplectic map" for ds/dt without defining it properly. Later in l. 93 suddenly canonical transformations are mentioned as being “symplectic maps” without explaining why. Since these notions are at the core of Sym measure, they need to be defined clearly so they can be taken up again when introducing the measure. Please clarify.
- The Sym measure requires training a neural network for converting the learned latent state into the ground truth state representation. This is problematic for two reasons: a) the proposed measure is only of theoretical relevance, because it cannot be used in practice when ground truth is not available (otherwise learning a latent state representation would also be useless and the latent state mapping of the images could be directly learned onto the ground truth state representation). b) The analysis depends on the quality of the learned mapping which is unclear and can be prone to model errors.
- Eq 2 has a sum ranging over t but it’s not used for any term inside the sum. Above the equation, the distribution p_\theta(x_t) = d_\theta( q_t ) is mentioned but never used again. What is the argument q_t of d_\theta ? What is p_\theta( x | s ) in eq 2 ?
- The paper lists improvements over HGN but does not give insights why these changes make the model better.
- l. 119, how can a 1D phase space make sense, shouldn’t it be at least 2D (1D position and 1D momentum)? Can a 1D phase space represent all the environments which are evaluated in Sec. 7?
- l. 122 what does it mean to train the network on a “prediction” rather than a “reconstruction” task?
- Eq. 3 what is the definition of \hat{s} ?
- l. 224. The R^2 measure is motivated as goodness of fit, but it does not follow the usual definition \sum_i (F(S_i)-E(s_i))/E(s_i). Instead, it seems to compare the variance of mapping F(S) with the variance of the ground truth. The paper does not explain clearly, why this should be a meaningful measure. Why is a low number indicative of bad model performance? What happens if the learned mapping predicts larger or lower variance than the ground truth ? The motivation of thresholding for a low R^2 in l. 262 cannot be followed.
- The proposed SyMetric measure is an ad-hoc combination of the R^2 and the Sym measure and relies on an empirical choice of two thresholds. What does it mean for the thresholds to “work well”? How to choose them subjectively?
- What do the colors of the curves in the B plots mean? If the curves are prediction vs ground truth it seems the learned mass-spring +c clearly gets out of sync which is also detected by VPT, while the proposed SyMetric measure wrongly finds the learned model to mimic the ground truth dynamics well. Please discuss.
Overall the technical description of the proposed measures seems flawed which make the method difficult to comprehend. The experiments do show issues of the proposed metric where the baseline VPT method seems sufficient and superior. The paper does not seem ready for publication yet.



Further comments:
- The description of HGN in sec. 3 should cite the original paper [48]
- def of VPT: please use “\operatorname{arg\,min}”

== Post-rebuttal comments ==

The author response addresses several of my concerns well. I still rate the paper borderline ("marginally above acceptance threshold"), since the approach requires to learn a mapping from the learnt latent state space to ground truth which could be difficult to assess in its quality itself and can invalidate the proposed measures if the mapping is "off". The paper should provide an adequate assessment of this potential problem, for instance, suggesting ways to analyze the accuracy of the learned mapping and its impact on the proposed measure.

**Time Spent Reviewing:**

8

---

> ### Author Response · Authors · 2021-08-10
> **Response - technical description**
>
> We thank the reviewer for their comments. Our understanding is that the reviewer’s main concerns with the paper can be summarised as: ”(1) Overall the technical description of the proposed measures seems flawed which make the method difficult to comprehend. (2) The experiments do show issues of the proposed metric where the baseline VPT method seems sufficient and superior.” We will address these in turn.
>
> (1). The connection between symplectic maps, canonical transformations and Hamiltonian dynamics can be best described as the following: Symplectic maps and canonical transformations are two terms coming respectively from mathematics and physics to describe the same thing - the flow of a Hamiltonian vector field (see e.g. Abraham and Marsden, Foundations of Mechanics, 1978). We did not expand on the details too much in the paper due to the lack of space and the large availability of relevant material online. Saying this, we are happy to include more references and a more explicit statement linking all three terms together in the updated version of the manuscript. We will also expand the ds/dt equation in Supplementary Materials for more clarity as in Section 16.3 in https://webspace.science.uu.nl/~frank011/Classes/numwisk/ch16.pdf.
>
> The reviewer also mentioned a concern that the measure “is of only theoretical relevance, because it cannot be used in practice when ground truth is not available (otherwise learning a latent state representation would also be useless and the latent state mapping of the images could be directly learned onto the ground truth state representation)”. Although our measure does rely on having the ground truth state information, it can be calculated using significantly fewer ground-truth labelled data points than would be necessary to train any sort of NN if it were to learn the dynamics from the state directly. As an illustration, the original HNN [18] required 8,000 training points to learn two-body dynamics, while the polynomial version of our metric can be calculated using 600 data points, a more than 10x reduction in the quantity of the needed data. Training on large scale data and evaluating on small scale labeled data has become standard practice in other areas in machine learning. For example, quantitative evaluation of language models and self-supervised methods in vision follows this exact protocol (e.g. Devlin et al, 2019 in Tbl. 1; Chen et al, 2020 in Tbl. 8). We believe that it is feasible for the practitioners to collect such relatively small amounts of ground truth information for a small subset of trajectories in many domains (either experimentally or through simulation).
>
> Furthermore, the methods that learn Hamiltonian dynamics from high dimensional observations [48,8,44,2,57], which our measures are aimed at, were developed for a particular application in mind -  inferring the dynamics from high-dimensional observations at test time. This is important in robotics, self-driving cars and other RL applications. For these use cases, one still needs to learn from pixel observations regardless of how much ground truth labelled data is available. The question then is - how do you know that your model actually learnt the right dynamics? How can you have confidence to use your model at test time for inference? Currently there is no way to measure this, and this is the problem we tackled in our paper.
>
> The reviewer also raised a concern that “the analysis depends on the quality of the learned mapping which is unclear and can be prone to model errors.” Many other metrics routinely used in ML also rely on the quality of a learnt mapping (e.g. many of the metrics for evaluating generative models, like GANs, see Goodfellow et al, 2014; Salimans et al, 2016, Heusel et al, 2017; many of the disentanglement metrics like Eastwood and Williams, 2018; Chen et al, 2018; Duan et al, 2020; even the scoring functions for AlphaFold, Senior et al, 2021). We believe that as a ML community, we are well equipped to detect and deal with any potential model fitting problems, and hence this is not a limitation of the approach.
>
> The reviewer also questions our choice of the R^2 measure vs  \sum_i (F(S_i)-E(s_i))/E(s_i). R^2 is commonly used in statistics to measure how well the model explains the data (see e.g. Wright, 1921). It is a better measure than error based measures (e.g. MSE) because it is independent of the magnitudes of the variables, and hence the threshold can be meaningfully set across all datasets. We did try MSE instead of R^2 during the development of the measures, which worked equally well, and we are happy to add those results to the updated version of the paper.
>
> To address the reviewer’s question on combining R^2 and Sym scores into a single binary indicator. As mentioned in the paper, a model has learnt the dynamics well if the latent dynamics are both informative of the ground truth latent dynamics (as measured by R^2) and the mapping between the two is symplectic (as measured by Sym). Hence, in the paper we suggest that practitioners use both of these scalars to understand how well their models are doing. Indeed, these two measures are shown to correlate better with the model success than the MSE measures of similar data efficiency (see lines 286-294). The binary combination of the two (and the corresponding thresholds) is our way to also give users a shorthand for a faster, "at a glance" kind of understanding of how well their models are performing. This is not meant to be prescriptive, since, as the reviewer has correctly pointed out, the thresholds have to be chosen subjectively (although the results are quite robust to the precise choices), and the users are encouraged to use R^2 and Sym for a more in depth understanding of their models. We will make this point clearer in the paper.

---

> > ### Author Response · Authors · 2021-08-10
> > **Response - experiments**
> >
> > (2). The colours in Fig.2 B are indeed prediction (blue) vs ground truth (orange) trajectories (we will update the legend). The two trajectories do get out of sync with each other on the mass-spring +c dataset, however this is not because the model has learnt the wrong dynamics. The dynamics are the same since the two curves are the same up to a phase shift. Instead, the curves go out of sync because of the error in the inferred initial state. This is likely due to the fact that the images are rendered at a low resolution, and aliasing can lead to small errors in the inferred initial state that would result in the observed phase shift. Given that we are primarily interested in measuring the quality of the learnt dynamics rather than the quality of the inference network (although of course that is also important, but it is not what our measures are meant to detect), SyMetric is in fact more accurate than VPT.
> >
> > We hope that our answers above resolve any misunderstandings that may have arisen and demonstrate that our technical description of the proposed measures is not flawed. We are not sure which experiments the reviewer refers to when claiming that the baseline VPT measure is superior - every result in the paper suggests that VPT is either equal or worse than our proposed measures (Sym and R^2 and their combined SyMetric) while requiring almost 20x longer trajectories, which may not be feasible to obtain in many domains of interest.

---

> > > ### Author Response · Authors · 2021-08-10
> > > **Response - minor comments**
> > >
> > > The other minor comments are addressed below:
> > >  - "Eq 2 has a sum ranging over t but it’s not used for any term inside the sum. What is the argument q_t of d_\theta ? What is p_\theta( x | s ) in eq 2?"
> > >
> > > We thank the reviewer for pointing out our oversight. All variables x and s in the equation should have the time subscript t. We will add those in. q_t of d_\theta is the position component q of the full latent state s=(q,p) at time t that is used to reconstruct the observation x at time t. p_\theta(x|s)  is the probability of the observation x conditioned on state s. It is estimated by the model decoder with parameters theta that takes in the positional variables q of state s=(q,p) and produces the observation x.
> > >
> > > - "l. 119, how can a 1D phase space make sense, shouldn’t it be at least 2D (1D position and 1D momentum)? Can a 1D phase space represent all the environments which are evaluated in Sec. 7?"
> > >
> > > The reviewer is correct that the phase space is always in R^{2N}, to account for the matching position (q in R^N) and momentum (p in R^N) coordinates. What we mean by 1D and 2D in this paragraph, however, is whether the latent space is represented as a spatial feature map (2D) - being represented as an HxWxD array, as in the HGN paper [48] - or is represented as a single vector in R^D (1D). We always initialise our latent space to have more dimensions than the ground truth state space, so it is always capable of representing all the environments.
> > >
> > > - "l. 122 what does it mean to train the network on a “prediction” rather than a “reconstruction” task?"
> > >
> > > In the original HGN, the network is presented with N observations [x_1, ..., x_N]. It then infers the initial state corresponding to the first time step s_1 and produces a rollout up to time step N. This rollout is then decoded to produce [\hat{x}_1, ..., \hat{x}_N], and the MSE(x, \hat{x}) is calculated between the ground truth observations and the "reconstructed" input observations. In HGN++ we instead use the observations to infer the state s_N at time N and then produce a rollout T steps into the future with respect to the observed images  [\hat{x}_N, ..., \hat{x}_N+T], hence producing a "prediction" of future observations.
> > >
> > > - "The paper lists improvements over HGN but does not give insights why these changes make the model better."
> > >
> > > We believe that the modifications that make up HGN++ help with the optimization of the HGN learning objective. For example, the swish non-linearity has better differentiability properties when it comes to higher order derivatives that are required for optimizing the Hamiltonian function approximator; the 1D phase space is more general than a 2D latent space, which builds in an inductive bias on the spatial properties of the problem that may not apply universally; and training the network for “prediction” rather than “reconstruction” is more directly related to how the models should generalise at test time. We will include a discussion of these points in the updated manuscript.
> > >
> > > - "Eq. 3 what is the definition of \hat{s}?"
> > >
> > > Thank you for spotting our omission. \hat{s} is a ​​projection of the learnt state space S back to the ground truth state space $\hat{s}(t) = F(S(t))$
> > >
> > > - "Further comments: 1) The description of HGN in sec. 3 should cite the original paper [48]; 2) def of VPT: please use “\operatorname{arg,min}”
> > >
> > > We will make the changes.

---

> > > > ### Author Response · Authors · 2021-08-31
> > > > **Thank you for considering our response**
> > > >
> > > > Dear Reviewer,
> > > >
> > > > Thank you for taking the time to consider our response and for increasing your score. We really appreciate it.
> > > >
> > > > We completely agree that it is very important to learn a good mapping between the learnt latent state space and the ground truth state, and that if the mapping is "off", it will affect the interpretation of the Sym score. Indeed, if the quality of the mapping is bad, Sym scores are meaningless. This is why we suggest using Rsq as a way to quantify the quality of the mapping in a way that avoids the particularities of individual datasets (unlike MSE or other pixel based measures). We then prescribe to only consider Sym scores if Rsq is high. We will make sure to make this point more explicit in the updated version of the paper.

---

### Author Response · Authors · 2021-08-10
**Comment to all reviewers**

Dear Reviewers,

We thought we would clarify once again the problem that our paper is trying to address. There is currently an emerging subfield in the community that tries to incorporate priors from classical mechanics to learn dynamics from high-dimensional observations, like pixels, rather than low-dimensional state [48,8,44,2,57]. These models are designed to infer the underlying dynamics faithfully from such pixel observations at test time, with applications in robotics, self-driving cars and other RL problems. Before deploying such models, it is important to be confident that they have learnt the underlying dynamics well. However, currently no method exists to do this. Not only that, as we show in the paper, the reconstruction-based measures used in the community as proxies of the quality of the learnt dynamics are misleading. In this area, this is an important problem to address, and our paper is the first to do so.

Our contributions are:
1) The first comprehensive demonstration of the flaws of the current pixel-based evaluation methods. These methods dangerously overestimate the ability of the models to learn dynamics.
2) The first principled method for measuring the quality of the learnt dynamics directly - a set of two scalar measures that together give practitioners a better understanding of how well their models have captured the underlying dynamics, and a recipe for how to combine them for a “quick glance” indicator of which models are doing better than others.
3) A comprehensive demonstration that our proposed measures are more accurate than the traditional pixel-based measures. This is supported by a
- quantitative confirmation in terms of consistently improved model performance when using our measures as a signal for model selection during hyperparameter search.
- qualitative analysis visualising the model trajectories both in pixel and latent space extrapolated over 100x more of steps that used for training the models, and the first demonstration of  semantic interpretability of the learnt latent space in this model class;

---

### Decision · Program_Chairs · 2021-09-27

**Decision:**

Accept (Poster)

**Comment:**

This paper has three reviewers, and the initial review are two borderline reject, and one borderline accept. After very long and detailed rebuttal and discussion process between authours and reviewers, two reviewers had raised their scores. In general, this paper has a theoretically grounded working solution to a valid and important problem in the field. The negative points/drawbacks come from that the results are not convincing and partially negative results which are very hard to impossible to support empirically. And the  approach needs to learn a mapping from the learnt state space to ground truth which is difficult to assess in quality itself and can invalidate the proposed measures if the mapping is "off". Thus, the initial meta reviewer would tend to borderline acceptance.